cognition

colaughter, affiliation, social signalling, conversation, vocal communication

**Author for correspondence:**
Gregory A. Bryant
e-mail: gabryant@ucla.edu

# Recognizing affiliation in colaughter and cospeech

Gregory A. Bryant[1,2], Christine S. Wang[1]
and Riccardo Fusaroli[3,4]

[1]Department of Communication, and [2]Center for Behavior, Evolution, and Culture, University of California, Los Angeles, CA, USA
[3]Department of Communication and Culture, and [4]Interacting Minds Center, Aarhus University Denmark

GAB, 0000-0002-7240-4026; CSW, 0000-0002-7801-4299;
RF, 0000-0003-4775-5219

Theories of vocal signalling in humans typically only consider communication within the interactive group and ignore intergroup dynamics. Recent work has found that colaughter generated between pairs of people in conversation can afford accurate judgements of affiliation across widely disparate cultures, and the acoustic features that listeners use to make these judgements are linked to speaker arousal. But to what extent does colaughter inform third party listeners beyond other dynamic information between interlocutors such as overlapping talk? We presented listeners with short segments (1–3 s) of colaughter and simultaneous speech (i.e. cospeech) taken from natural conversations between established friends and newly acquainted strangers. Participants judged whether the pairs of interactants in the segments were friends or strangers. Colaughter afforded more accurate judgements of affiliation than did cospeech, despite cospeech being over twice in duration relative to colaughter on average. Sped-up versions of colaughter and cospeech (proxies of speaker arousal) did not improve accuracy for either identifying friends or strangers, but faster versions of both modes increased the likelihood of tokens being judged as being between friends. Overall, results are consistent with research showing that laughter is well suited to transmit rich information about social relationships to third party overhearers—a signal that works between, and not just within conversational groups.

## 1. Introduction

During social interactions, people produce a variety of dynamic behaviours that not only serve to function within an interacting group [1], but can also inform third parties about the nature of their social relationships and their intentions in a broad sense. Imagine encountering a group of people talking and suddenly everybody in that group erupts in spontaneous laughter. There

are many inferences an overhearer might draw about the people laughing together, including their history, current emotional states, shared information and even the interpersonal affiliation between specific dyads within a larger collection of laughers. To what extent does a group laughing together provide information about their relationship beyond just hearing them speak with one another? How well can overhearers generate accurate inferences, what information is offered, and how do acoustic features of the laughter itself play a role? Here, we explore the relative roles of laughter and talk in intergroup communication.

Human laughter is characterized by a series of rapid bursts of vocal energy, often called bouts [2] that occur primarily in conversational interactions [3–5]. The initial burst is typically loudest with successive bursts often decaying in both frequency and amplitude [6]. Laughs can be either voiced or unvoiced, with voiced laughter following rather simple vowel production rules [7,8]. Laughter is highly recognizable across widely disparate societies and has proven to be a very robust indicator of positive emotion [9,10]. But by contrast, it is also associated with social ostracism and humiliation [11–13]. Evidence for highly distinct laugh types related to positive and negative emotions is limited, with some research suggesting distinctions across social functional categories (e.g. [14]), and other work pointing to the importance of context, with a high level of ambiguity in the physical properties of different laughs related to affect (e.g. [15]).

The complex pragmatic utility of human laughter is not well understood. A fairly substantial body of work strongly suggests that laughter functions, at least in part, to communicate positive affect and cooperative intentions among groups of mutually trusting individuals in ongoing relationships. The notion that laughter helps people signal affiliative intentions is central to all current functional approaches [13,15–26]. But few theorists have considered seriously the ubiquitous phenomenon of laughter in groups. Dezecache & Dunbar [20] provided an interesting exception in their ethological analysis of the typical sizes of natural conversational and laughter groups, which were similar, and generally around three or four individuals. They proposed a 'grooming at a distance' hypothesis, explaining group laughter (and conversation) as a means to regulate cooperative relationships beyond the limit of one imposed by physical grooming. When people laugh in groups they often do so together, but limited work has explored how people laughing together affects perceivers or what information it might contain.

The focus of the current study is the perception of colaughter. Here we define colaughter as temporally coincident laughter between two or more individuals who are in close spatial proximity to one another and engaged in shared attention. People often indicate affiliation when colaughing, and sometimes direct it negatively, either in groups or alone, toward those with whom they do not wish to affiliate. Colaughter production (also called coactive, shared, reciprocal and antiphonal laughter) arises early in development [27] and promotes affiliative feelings, higher perceived personal similarity, and reports of subsequent increases in relationship satisfaction and intimacy in those who produce it [28–30]. Additionally, colaughter has been associated with mutual sexual interest in brief cross-sex encounters [31], cognitive similarity [32], and the behaviour is notably different between men and women, as well as between friends and strangers. For example, in one study, female friends reported engaging in shared laughter earlier in their friendships than male friends (three weeks versus six weeks) [33]. Bryant [34] found that in recorded conversations, female friends generated significantly more frequent colaughter than cross-sex friends, or male friends, and friends in general produced higher rates of colaughter than strangers. Acoustically, colaughter was louder than individual laughs produced by the same speakers, and colaughter between friends was louder and had greater pitch variability than colaughter between strangers. Overall, a variety of research findings suggest that colaughter is rich with social information for overhearers.

There are clear benefits for individuals to develop sensitivity to subtle indicators of alliance structure, as well as benefits for allied signallers to convey that information reliably [35]. If these positive pay-offs held consistently for typical group interactions, social vocalizations such as colaughter that reliably correlated with cooperative relationships could have evolved for intergroup purposes. This social group signalling approach extends the functions of laughter beyond the immediate interactive context and, as explained below, can help elucidate some of laughter's more unique acoustic and psychological features. Non-human animal examples of group signalling are abundant—animals collectively signal group territory boundaries, mateships and other identifying information by chorusing for other groups and individuals. Humans exhibit many group-level vocal signals as well, including ritualized chanting, group singing, coscreaming, among others—signals that broadcast to those outside the group information about identity, intentions and coalition strength [36–38]. Could colaughter function similarly?

From a signal design perspective (i.e. how signals are shaped by natural selection to solve adaptive problems of communication), laughing is well suited for intergroup communication. Laughter has

acoustic features that are traditionally associated with wide broadcast and the penetration of noisy environments [39]. First, laughs often contain alerting components, such as high energy voice-onsets (e.g. high pitch and loudness with abrupt onset) effective for grabbing attention [40,41]. Second, laughter is conspicuous—it contains fairly distinctive acoustic attributes that differentiate it from most other vocal signals (crying is an interesting exception with several similar acoustic characteristics). Third, laughter comprises small repertoires—a reasonably stereotyped form resulting from automatic and rhythmic neuromuscular oscillations. Fourth, laughter is typically repetitious. A single laugh bout with simple acoustic elements can last for many seconds, and laugh epidemics have been documented with laugh episodes continuing for hours and even days [42]. Finally, laughter is highly contagious [43], pointing to design for group production. This collection of physical characteristics makes human laughter fairly exceptional across the known variety of play vocalizations in social mammals, most of which are difficult to hear at even a small distance. Human spontaneous colaughter can be construed as a derived activity originating from the co-production of play vocalizations [44], with the features described above resulting from ritualization—the process of a by-product cue becoming physically modified into a functional signal [45]. A form-function account that predicts and explains these acoustic characteristics is needed.

Recent research suggests that dyadic colaughter might be particularly effective in signalling information about interpersonal relationships. In a study examining the detection of friends versus strangers from short isolated clips of colaughter, listeners from 24 societies, ranging from small-scale hunter gatherers to industrialized college students, were able to reliably judge whether two people laughing together were friends or strangers [18]. Moreover, participants from different societies tended to use the same acoustic information to identify friends, specifically arousal-linked laughter elements such as shorter burst duration and irregularities in pitch and intensity cycles.

Sensitivity to the social relevance of laughter appears to develop early. Children begin laughing as early as six weeks, and before long, social events will trigger it [46]. Laughter is quite frequent in young children, but somehow it has escaped extensive empirical scrutiny in the developmental literature, a situation that is now changing [47]. Using a looking time paradigm, Vouloumanos & Bryant [48] found that five-month-old infants preferred colaughter between friends over that of strangers. A second group of five-month-olds were surprised when the source of the colaughter (friends or strangers) was incongruent with a video sequence of two people displaying either affiliative or non-affiliative behaviour. Another recent study documented the social nature of laughter production in preschoolers, showing that children as young as 3 years old produced up to eight times more laughter at funny cartoons when in the presence of at least one other child [49]. Larger groups did not elicit more laughter, and the amount of laughter was not associated with subjective levels of funniness of the stimuli in the children. This work conceptually replicated previous research done with 7 year olds [50]. The mere co-presence of others is enough to inspire the behaviour, suggestive of a developmental programme calibrating colaughter social signalling.

Adults and children are attuned to the acoustic features of colaughter between friends, with a likely attentional focus on speaker affect. Perceptually, one possibility is that high arousal is associated with spontaneous laughter [19,51] which, in turn, is judged as more likely to be produced between friends (see SI in [18]). Relative to volitional laughs, spontaneous laughs tend to be higher in pitch, higher in rates of intervoicing intervals (i.e. more time between voiced bursts), more irregular in pitch and intensity cycles, and noisier (for a review see [17]). In a large cross-cultural study, participants from 21 societies reliably distinguished spontaneous laughter from volitional laughter, again relying on related arousal-linked acoustic phenomena to make their judgements as listeners identifying friends in colaughter across cultures [9].

Other types of spontaneous emotional vocalizations differ from their volitional counterparts on similar acoustic features tied to speaker arousal, such as raised and more variable fundamental frequency, and lower harmonicity [52]. Sensitivity to indicators of speaker arousal could help overhearers track emotional engagement between interactants. This provides the basis for possible subsequent positive selection on groups of senders to amplify the signal, resulting in a ritualization process [44]. Proximately, this can manifest itself in familiar group members as heightened experiences of mirth and joy that motivates further colaughter. Moreover, there could be additional elements of group interactions that contribute to people's affective experiences and subsequent colaughter features. For example, reduced behavioural inhibition and increased overall comfort between familiar speakers could afford colaughing episodes that are not necessarily reflecting greater arousal, but more honest sounding vocal emotions and relaxed engagement. While recent studies strongly suggest arousal-linked qualities are playing a role in people's judgements of familiarity, affiliation is potentially revealed in a variety of ways through interactive dynamics.

In summary, there are now well-documented, wide cross-cultural consistencies in the perception of affiliation in colaughter, as well as the perception of laughter as spontaneous or volitional. These results provide preliminary but compelling evidence that laughter has recognizable universal structural aspects that communicate important information about affect, relationships and intentions.

Here, we investigated the relative efficacy of laughter to communicate information about social affiliation by comparing colaughter to overlapping speech (cospeech). We consider cospeech to be a useful co-interactive behavioural control to make this comparison. As in laughter, there are many reasons that people overlap in their talk, including backchannelling (i.e. brief utterances while listening in conversation), interruptions, and ordinary turn-taking in which the vocalized offset of one speaker minimally overlaps with the onset of an interlocutor in fairly precise ways [53]. These conversational phenomena can be associated with a variety of pragmatic effects [54]. Perhaps laughter is not special in its ability to reveal affiliation, and instead the dynamics of other vocal interactions can reveal equally reliable information to overhearers. Thus, we used cospeech here to act as an appropriate baseline behaviour in conversational interaction rather than a viable alternative as a potential signal of affiliative status between speakers. Emotional signals such as colaughter contain rich information about the mutual affective intentions between socially interacting individuals, including honest signals of the physiological states of the interactants. Colaughter between friends is more likely to be judged as containing individual spontaneous laughter [9,18], a product of the evolutionarily conserved vocal emotion system [55–57]. These signals can reveal social information in a way that ordinary speech typically does not. By this logic, we expected that colaughter would be more effective in conveying speaker affiliation than cospeech.

Additionally, we explored whether vocal production speed—an arousal-linked feature present in both laughter and speech—would affect judgements of affiliative status. Again, because laughter is a carrier signal of both physiological arousal and emotional valence, we expected that increased laughter burst rate—an index of production speed in laughter—would increase judgements of affiliation. But an equivalent increase in speech rate (i.e. syllables per second) would not result in increased judgements of affiliation in cospeech by the same speakers. While arousal can have perceptible effects on speech rate [58], there are many non-affective reasons that people alter their speech rate, and intra- and interspeaker variability are high.

# 2. Method

Using edited clips from conversational interactions between friends and strangers described below, we presented listeners with interspersed recordings of colaughter and cospeech, hearing either a sped-up version or the original version of any given token. Listeners were asked to identify whether the pair of vocalizers were friends or strangers and were additionally asked to rate how well they believed the interactants liked one another.

## 2.1. Participants

We tested 108 participants (28 male) (mean age = 19.2; range = 17–23; s.d. = 0.9) who completed the experiment for credit in an introductory communication course at University of California, Los Angeles. The experiment was approved by the UCLA Institutional Review Board, and all participants provided informed consent prior to participating.

## 2.2. Materials and procedure

### 2.2.1. Laughter and speech stimuli

All stimuli were extracted from 24 recorded conversations between either established friends (mean length of acquaintance = 20.5 months; range = 4–54 months) or newly acquainted strangers who met immediately prior to the recording. All conversationalists were college-aged students (friends mean age = 18.6, s.d. = 0.6; strangers mean age = 19.3, s.d. = 1.8) who were participating in the recording session for credit in an undergraduate psychology course. Conversations were recorded digitally (Sony DTC series DAT recorder, 16-bit, 44.1 kHz) in a quiet room using lapel microphones (Sony ECM-77B) in the Fox Tree laboratory at UC Santa Cruz in 2003. Speakers were initially instructed to

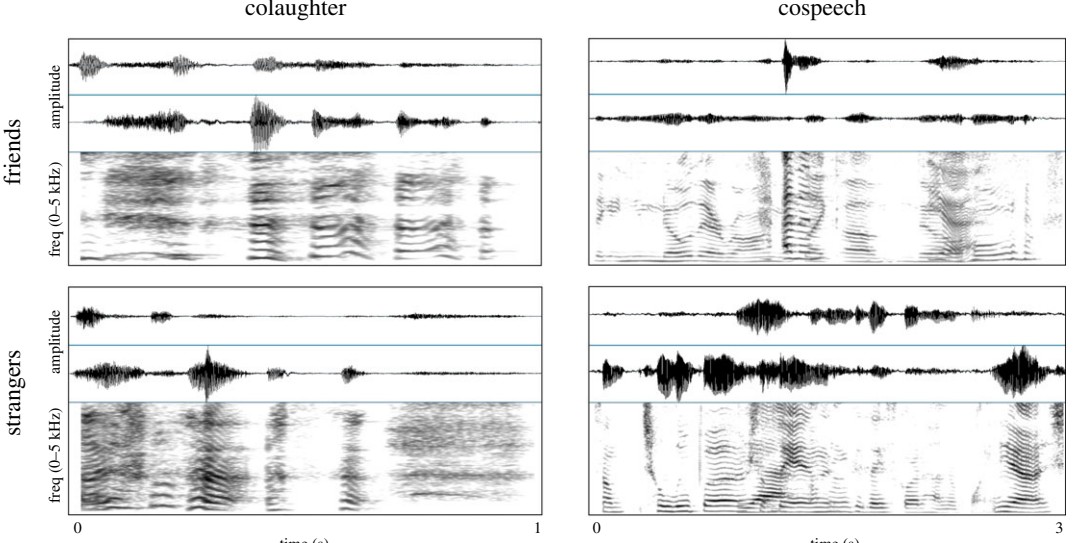

**Figure 1.** Sample waveforms and narrowband FFT spectrograms (30 ms Gaussian analysis window, 44.1 kHz sampling rate, 0 to 5 kHz frequency range) of colaughter and cospeech by friends and strangers.

talk about bad roommate experiences, but were told they could talk about any topic. Conversations lasted for an average of 13.5 min (s.d. = 151.3 s). For more details of the conversations, see Bryant [59].

The original set of 48 colaughter segments have been used in two previous studies [18,48], with detailed descriptions available, including acoustic information. In sum, colaughter was defined as simultaneous (offset of an initial laugh within 1 s of onset of a subsequent laugh) laughter produced by two speakers without verbal or other audible sounds present. From 24 conversations (12 between friends and 12 between strangers), the first and last occurrences of qualifying bouts of colaughter were used for a set of 48. Between friends and strangers, colaughter segments were not different in duration ($M = 1.1$ s; s.d. = 0.37 s) or laughter onset asynchrony ($M = 313$ ms; s.d. = 257 ms).

Cospeech was defined as simultaneous speech production (energy onset within 1 s) between two speakers. The first and last instances of cospeech were excised from the same conversations as the colaughter described above, providing a set of 48 cospeech clips. These initial criteria were the same as those used to extract colaughter samples. But cospeech, of course, differs from colaughter on many dimensions, including its likely greater heterogeneity in pragmatic functioning. Additional criteria were used for choosing cospeech samples that probably increased their coherency, and thus their potential ability to reveal relationship dynamics. Instances had to contain no greater than a 4 : 1 ratio of speech time between speakers, no laughter, no successful interruptions and no verbal information that would identify the speakers as friends or strangers. The average duration of cospeech samples was 2.3 s (s.d. = 0.93), which is greater than two times the length of the corresponding colaughter samples. Speech rates (measured as syllables per second) of cospeech between friends ($M = 3.7$, s.d. = 1.6) and strangers ($M = 3.9$, s.d. = 1.75) were similar, $t_{94} = 0.67$, $p = 0.51$. Pilot work in developing these stimuli revealed that clips of cospeech matched in duration to the corresponding colaughter clips from the same interactants (approx. 1 s) did not afford accurate judgements of friends versus strangers, providing initial support for the expectation that laughter provides richer information about affiliation than does speech. To confirm that the verbal information alone would not reveal the relationship status, cospeech samples in the current study were transcribed verbatim and the text was presented randomly on a computer (iMac; SuperLab 4.0) to a separate group of participants from the same pool as the main experiment. Participants were equally likely to judge the text as being between friends for both familiar ($M = 0.61$, s.d. = 0.13) and unfamiliar dyads ($M = 0.59$, s.d. = 0.14), $t_{64} = 0.53$, $p = 0.60$. For the entire text of the cospeech stimuli, see electronic supplementary material, table S3.

From the complete set of 96 tokens, we created a speed-manipulated set. Samples were sped-up (duration reduced 33%) with pitch held constant using the Adobe Audition 2.0 (www.adobe.com) constant stretch effect function (stretching mode: time stretch, high precision, splicing frequency: 51 Hz, overlapping: 30%, ratio = 150). All tokens were normalized to peak amplitude. See figure 1 for spectrogram examples of colaughter and cospeech for both friends and strangers. See electronic supplementary material for one audio example from each condition.

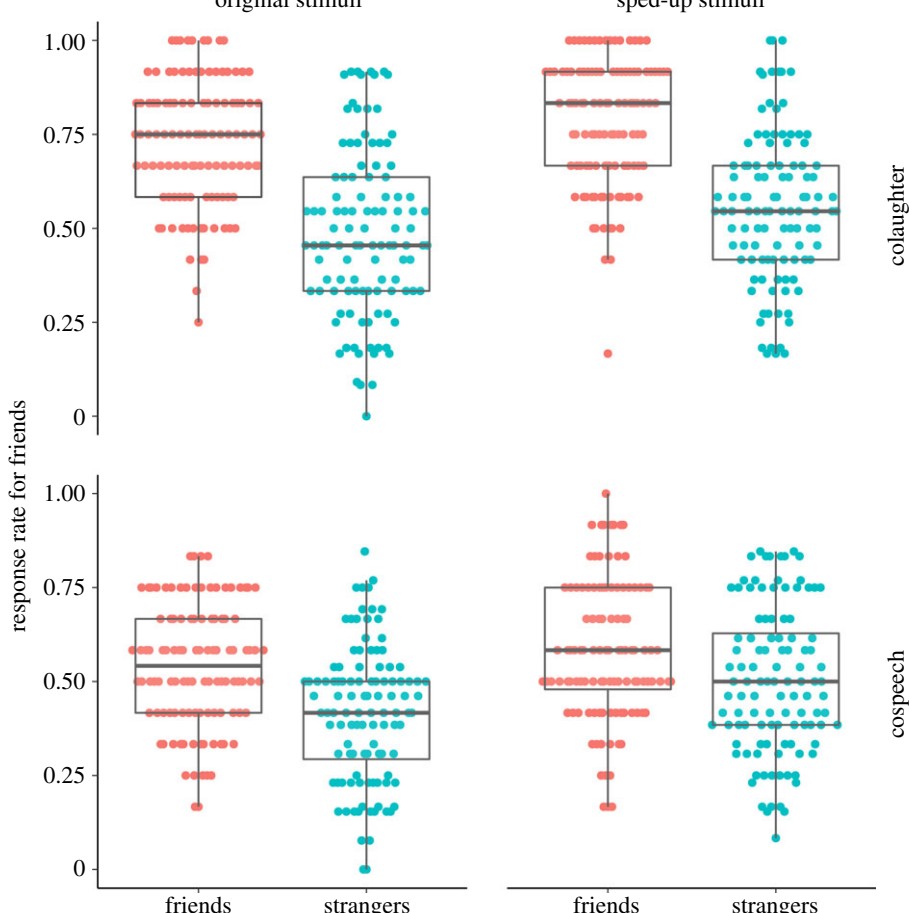

**Figure 2.** Response rates for answering 'friends' in judgement task across conditions of actual familiarity, mode of vocal production and vocal production speed. Each point represents the average rating across 12 exposures to the condition for each participant (i.e. 108 data points per condition).

## 2.2.2. Procedure

Two stimulus lists were created with half of the manipulated colaughter and cospeech clips in each list. Participants were then presented one of the two lists, thus only hearing one version of each clip (96 trials).

The experiment was presented using SuperLab 4.0 (www.superlab.com) on an iMac desktop computer in an experimental cubicle in a quiet room. Participants wore headphones (Sony MDR-V250) and loudness levels were checked prior to each session. The 96 colaughter and cospeech clips were presented in random order, and after each recording, participants were asked (i) decide whether the people interacting were friends or strangers by pressing either '0' for 'strangers' or '1' for 'friends' on a computer keyboard, and (ii) how much do you think these people liked each other on a scale of 1 to 7 where 1 is 'not at all,' 4 is 'somewhat' and 7 is 'very much'. Prior to beginning, participants were told that some of the pairs of people were friends at the time of the recording, and others were complete strangers who were meeting for the first time. After one practice trial, the experiment began. See electronic supplementary material for text of complete instructions.

Participants answered two questions after each trial. The first question was to identify whether the presented pair of speakers were friends or strangers. Raw response rates for answering 'friends' in the judgement task across the conditions of speaker familiarity, mode of communication and vocal production speed are presented in figure 2. The second question was 'How much do you think these people liked each other?' Results for this second question are presented in the electronic supplementary material.

## 2.2.3. Statistical modelling

We performed a signal detection analysis in the form of Bayesian multilevel probit regressions [60] with weakly regularizing priors. The binomial response (judgement of 'friends' versus 'strangers') was

predicted by intercept (equivalent to criterion, or response bias) and familiarity (familiar versus non-familiar dyads; equivalent to sensitivity or d prime). Criterion indexes participants' bias in judging stimuli as more or less likely to comprise friends independently from the actual category. The sensitivity indicates how well familiarity can be detected in the stimuli, controlling for the criterion (see Models 0 and 1). Further, we tested whether our experimental manipulations affected criterion and sensitivity by introducing main effects (relations to criterion) and interactions (relations to sensitivity) of cospeech versus colaughter, and speed with familiarity (see Models 2, 3 and 4).

All predictors were modelled as multilevel parameters (also called random effects); that is, varying by the participant, and as possibly correlated. Potential stimulus heterogeneity was also accounted for by varying the intercept by stimulus. Note that multilevel models perform partial pooling of information, so estimates of each participant and each stimulus are influenced by the data available for all participants. This might reduce differences between participants, but it also provides more conservative estimates and has been shown to improve the generalizability of the models [61].

This generated the following models:

Model 0: $\text{Probit}(\text{Response}_{ij}) = \alpha_{ij}$

Model 1: $\text{Probit}(\text{Response}_{ij}) = \alpha_{ij} + \beta_{1j} \text{ Familiarity}_{ij}$

Model 2: $\text{Probit}(\text{Response}_{ij}) = \alpha_{ij} + \beta_{1j} \text{ Familiarity}_{ij} + \beta_{2j} \text{ Talk}_{ij} + \beta_{3j} \text{ Familiarity}_{ij} \text{ Talk}_{ij}$

Model 3: $\text{Probit}(\text{Response}_{ij}) = \alpha_{ij} + \beta_{1j} \text{ Familiarity}_{ij} + \beta_{2j} \text{ Speed}_{ij} + \beta_{3j} \text{ Familiarity}_{ij} \text{ Speed}_{ij}$

Model 4: $\text{Probit}(\text{Response}_{ij}) = \alpha_{ij} + \beta_{1j} \text{ Familiarity}_{ij} + \beta_{2j} \text{ Talk}_{ij} + \beta_{3j} \text{ Speed}_{ij} + \beta_{4j} \text{ Familiarity}_{ij} \text{ Speed}_{ij} + \beta_{5j}$ $\text{Familiarity}_{ij} \text{ Talk}_{ij} + \beta_{6j} \text{ Talk}_{ij} \text{ Speed}_{ij} + \beta_{7j} \text{ Familiarity}_{ij} \text{ Talk}_{ij} \text{ Speed}_{ij}$

The $i$ subscript indicates 'for the $i$th stimulus' and the $j$ subscript indicates 'for participant $j$'. Thus, the $i$ and $j$ subscripts on the $\alpha$ coefficient indicate random intercepts by participant and stimulus, and the $j$ on a $\beta$ coefficient indicates a random slope for that predictor over participant.

We performed prior predictive checks to choose weakly regularizing priors for the model, excluding implausibly high values for the effects of the experimental manipulations while not critically affecting our results [62]. We tested normally distributed priors with a mean of zero and standard deviations of 1, 0.5, 0.3 and 0.1. We chose a standard deviation of 0.3 as the related predictive prior could generate a broader distribution of outcomes than our data and had a low probability for extreme rates of choosing 'friends'.

The models were run on two parallel chains with 3000 iterations each, an adapt delta of 0.99 and a tree-depth of 20 to ensure no divergence in the estimation process. Estimates from the models are reported as mean and 95% credibility intervals (CI) of the posterior estimates. We also report the credibility of the estimated parameter distribution: the probability that the true parameter value is above 0 if the mean estimate is positive, or below 0 if it is negative. The quality of the models was assessed by: (i) ensuring no divergences in the estimation process, (ii) visual inspection of the Markov chains to ensure stationarity and overlapping between chains, (iii) ensuring Rhat statistics to be approximately 1.00 and number of effective samples to be above 200, and (iv) comparing prior and posterior estimates to ensure the model was able to learn from the data. The relevance of the predictors was assessed first by model comparison relying on estimated out-of-sample error via Pareto-smoothed leave-one-out information criteria (LOOIC, [63]).

We also calculated the accuracy of judgements in terms of area under the curve (AUC) from the predictions of the above models. AUC is a robust measure of performance, accounting for baseline accuracy and the cost function of the judgement; that is, the possible relative weights put on the different kinds of errors (false positives and false negatives). To draw receiver operator characteristics (ROC) curves and calculate AUC performance, we estimated the predictions of the best signal detection model above and employed them to assess the effects of varying decision thresholds on the sensitivity and specificity of the model binomial predictions. To maintain intelligibility of the plots, we only represent ROC curves estimated from the means of the model posteriors. The results reported here are all calculated from the posterior estimates of the best model according to the model comparison procedure; that is, if the model included a three-way interaction, we calculated the actual contrast necessary to test our specific hypothesis and assessed its credibility instead of relying on the less specific three-way interaction.

All analyses were performed in RStudio 1.1.423, running R 3.5 and relying on the brms 2.40, pRoc 1.12.1 and Tidyverse 1.2.1 packages [64–67].

**Table 1.** Estimates of criterion, sensitivity and AUC (area under the curve—a measure of accuracy) for each condition, calculated from the posterior estimates in the signal detection model. Mean estimates for criterion and sensitivity are reported first on a percentage scale (0–100 percentage of choosing 'friends' when the stimulus is produced by strangers for criterion, difference in probability of choosing 'friends' between the stimulus being produced by friends and it being produced by strangers for sensitivity), then on a z-score scale (as in the SDT model, followed by 95% CI in parentheses (also on the original z-score scale). Area under the curve is on a 0–1 scale followed by 95% CI in parentheses and indicates the accuracy of the responses, with 0.5 being accuracy at chance level and 1 perfect accuracy. N = 108 × 96 trials (i.e. recordings) per participant = 10 368.

| condition | criterion | sensitivity | AUC |
|---|---|---|---|
| cospeech (original) | 41% 'friends' response, −0.22 (−0.3 −0.15) | + 13% in 'friends' responses, 0.33 (0.25 0.41) | 0.66 (0.64 0.68) |
| cospeech (sped-up) | 50% 'friends' response, −0.01 (−0.09 0.07) | + 9% in 'friends' responses, 0.22 (0.14 0.31) | 0.68 (0.66 0.70) |
| colaughter (original) | 49% 'friends' response, −0.03 (−0.13 0.06) | + 26% in 'friends' responses, 0.70 (0.62 0.79) | 0.75 (0.73 0.77) |
| colaughter (sped-up) | 56% 'friends' response, 0.16 (0.07 0.26) | + 24% in 'friends' responses, 0.69 (0.6 0.79) | 0.75 (0.73 0.77) |

## 3. Results

Model comparison indicated that the experimental conditions credibly affected the participants' responses (Model 4 having a lower estimated out-of-sample error than Models 0 to 3) (see electronic supplementary material, table S1). See figure 2 for raw response data on judgements of 'friends' in Question 1. Estimates from Model 4 are presented in table 1, and figures 3 and 4.

As predicted, participants were able to judge colaughter and cospeech above chance (AUC greater than 0.5, table 1 and figure 4), but were credibly better at judging colaughter—the difference in sensitivity between conditions was 0.37 (on a z-score scale), 95% CIs: 0.24 0.51, 100% credibility; that is, 100% of the estimated parameter values indicated a higher sensitivity for colaughter. In particular, colaughter was judged with 9% higher accuracy than cospeech. More exploratorily, we observed that participants were more likely to judge colaughter as produced by friends (62%) than cospeech (47.5%), a pattern that also held for stimuli produced by strangers only, with a false positive rate of 41% for cospeech and 49% for colaughter (mean difference in criterion between colaughter and cospeech: −0.19, 95% CIs: 0.07 0.32, 99.97% credibility). The overall pattern indicated that cospeech tended to generate more false negatives (i.e. cospeech produced by friends judged as produced by strangers) and colaughter more false positives (i.e. colaughter produced by strangers judged as produced by friends), but the total error rate was higher in cospeech.

As expected, speeding up colaughter increased the likelihood of it being judged as friends: 68% of 'friends' responses, against 62% in the original non-sped-up colaughter. The effect is due to participants developing a positive bias for colaughter produced by strangers: the 'friends' responses for the latter went from 49% to 56%, with an increase in the criterion of 0.19, 95% CIs: 0.09 0.29, 100% credibility. But their ability to judge colaughter produced by friends did not credibly change (going from 77% of 'friends' responses to 80%, mean difference in sensitivity between original and sped-up colaughter: −0.01, 95% CIs: −0.16 0.14, 44.97% credibility).

However, against our predictions, the effects of speeding up were not credibly different between colaughter and cospeech (mean difference in the criterion: −0.02, 95% CIs: −0.16 0.11, 63.6% credibility; mean difference in sensitivity: 0.1, 95% CIs: −0.12 0.31, 84.6% credibility). Speeding up cospeech affected participants' judgements by increasing their false positives rate from 41% to 50% (increase in the criterion for sped-up compared to original cospeech: 0.22, 95% CI 0.12 0.31, 100% credibility) and decreasing their sensitivity (−0.11, 95% CIs: −0.24 0.03, 94.73% credibility).

Detailed analyses of individual and stimulus variability in the effects are reported in the electronic supplementary material (i.e. see electronic supplementary material, figure S1–S4 and table S2).

Analyses reported in the electronic supplementary material on the ratings of liking (Question 2: How much do you think these people liked each other?) reflected the findings for judgements of friendship.

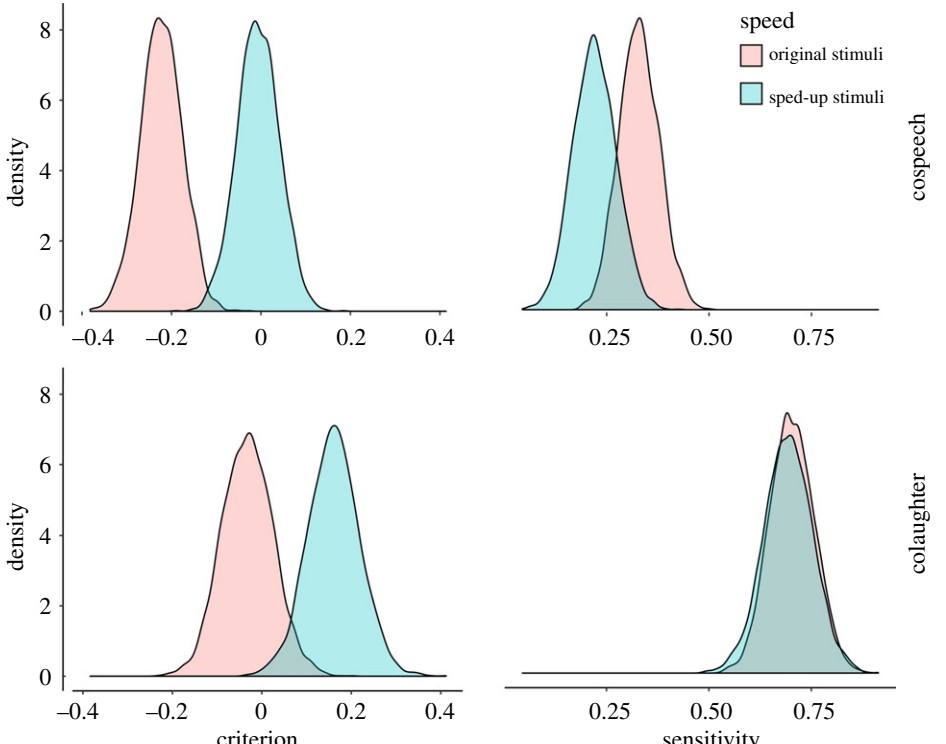

**Figure 3.** Posterior estimates of criterion and sensitivity from the Bayesian multilevel signal detection model. Negative bias indicates a tendency to respond 'stranger'.

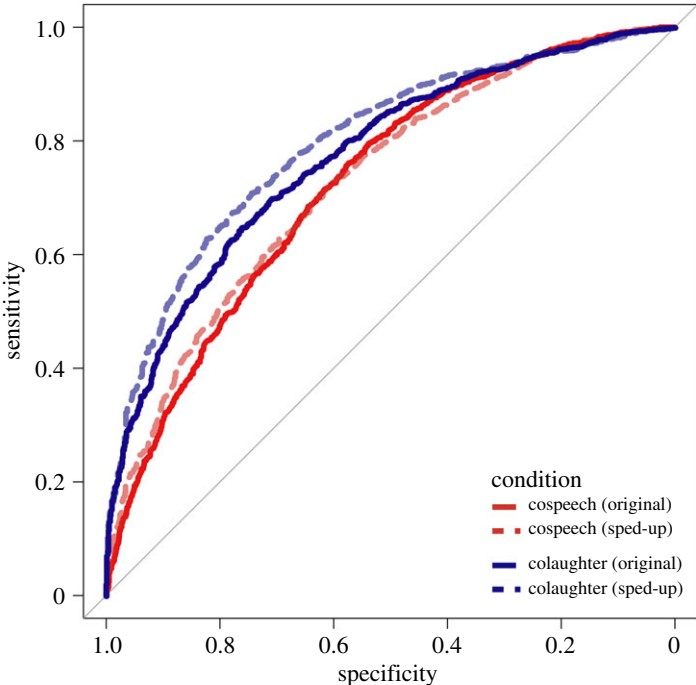

**Figure 4.** Receiver operator characteristics (ROC) for judgements of colaughter and cospeech in the original and sped-up versions. On the x-axis the proportion of strangers stimuli correctly identified, on the y-axis the proportion of friends stimuli correctly identified. The ROC is derived by generating predictions from the model and calculating their sensitivity and specificity as a function of the threshold for categorizing a stimulus as produced by 'friends.' In other words, we can set the threshold so that an estimated probability of 0.1 or more of being 'friends' leads to a 'friends' categorization, and estimate sensitivity and specificity. We can then increase the threshold to 0.2 and estimate sensitivity and specificity. This process (on a much finer scale) is then plotted generating the ROC curve.

They revealed that listeners gave higher ratings to: (i) actual friends over newly acquainted strangers, (ii) colaughter over cospeech, and (iii) faster colaughs and cospeech relative to slowed versions (see electronic supplementary material, figure S5 for rating data).

## 4. Discussion

Colaughter afforded more accurate judgements of friends and strangers than did cospeech, despite cospeech being over double in relative duration. This finding is consistent with the notion that colaughter constitutes an intergroup signal of affiliation to which listeners are highly sensitive. Given the ritualized, species-specific features of human laughter related to its apparent suitability for wide broadcast and signalling of affective information (e.g. colaughter is loud, conspicuous and contagious), it is not surprising that listeners were able to rapidly draw accurate inferences about the affiliative status of those laughing together. Cospeech, while revealing certain dynamics between interlocutors, is not well suited to rapidly and widely broadcast information, but instead is probably a by-product cue of conversational turn-taking and interpersonal coordination during discourse. As mentioned earlier, cospeech reflects a quite heterogeneous category of conversational linguistic phenomena given the complexity of language use and pragmatic signalling. Rather than thinking of cospeech as a communicative behaviour that can be contrasted with colaughter as a means of communicating affiliative behaviour, we instead consider it to be an appropriate baseline behaviour that affords an analysis of colaughter as a potentially specialized behaviour geared towards communicating affiliative relationships in group contexts.

An important signalling property of colaughter is its ability to efficiently transmit social information. Cospeech clearly also conveys social information to overhearers, but over much longer timescales. Laughter, like other emotional signals such as crying and fear screams, works well at short timescales and over relatively long distances. It is also the case that individual laughter is effective at rapidly conveying emotional intent between interlocutors. In fact, the effects of colaughter on overhearers are probably due to the characteristics of the individual laughs making up the colaughter, as synchrony in colaughter does not seem to affect listeners' judgements of affiliation, and ratings of the individual laughs on the dimensions of valence and arousal predict judgements of 'friends' in the same laughs when paired [18].

While our speed manipulation had an overall effect of causing listeners to judge sped-up covocalizations as more likely to be between friends, this was due to differential changes in response biases across laughter and speech. When cospeech was sped-up, negatively biased judgement patterns disappeared, possibly reflecting a sensitivity to arousal that might better characterize interacting friends. Conversely, sped-up colaughter introduced a positive bias in listeners, making them more likely to judge a colaugh segment as between friends. While our production speed manipulation differentially affected judgements in colaughter and cospeech, it did not do so exactly according to our predictions. Listeners are clearly drawing some inferences based on speech rate in cospeech related to the affiliation of speakers, and these inferences are probably tied to inferred arousal at some level. This is the first study, to our knowledge, that has examined whether overhearers can detect affiliation in overlapping talk—our results suggest that listeners can make this judgement, independent of verbal information, if they have at least two seconds of cospeech. The effect is not large, however, and in the context of multimodal information, might be easily obscured.

Theories of the social function of laughter always focus on pragmatic actions within the interacting group, and usually just the dyad. But recent research, including the current findings, suggests that its adaptive reach could quite easily extend beyond that. Laughter acoustic forms indicate a broadcasting function, and the high arousal common within strongly affiliative interactive groups can contribute to extremely loud and coordinated vocal outbursts. Moreover, laughter is highly contagious, making group colaughter ubiquitous across social contexts, providing chorusing utility. As stated earlier, there are many examples from the non-human animal behaviour literature documenting the coordinated signals between groups of animals that help them advertise different aspects of their social environment [37,38]. Humans' propensity for extensive cooperation beyond kin networks implicates a need for group-level signals that allow coordinated alliances to assess one another. Listeners' high sensitivity to the subtle dynamics of brief slices of colaughter—including preverbal infants and adults from all over the world—strongly suggests the existence of perceptual adaptations that track social alliances through vocal emotion behaviour.

The distinction between signals and cues is of paramount importance here. Many behaviours associated with actual cooperation appear to be subtle cues as opposed to robust signals. We believe the

evidence presented here reveals a signalling system—a system probably functioning for established friends, and not newly acquainted strangers. One of the primary distinctions between signals and cues is that in order to evolve, signals must benefit senders, whereas cues are given off inadvertently, and in many cases are the by-product of another system and difficult to hide [68]. The benefits are derived primarily by how the signals affect the behaviour of a target audience. In the case of individual spontaneous laughter, the signal often encourages continued interaction with the target and predicts future positive engagement, including cooperative interaction and support [13,19]. Laughter can act as a covert signal of shared information, often mediated through encrypted, intentional humour [69], affording adaptive strategies of social alignment through the mutual recognition of spontaneous co-signals within groups [32,70,71]. Broadcasting such a state of affairs to other groups could serve to influence how outsiders might interact with that group, which can include both encouraging interaction in some contexts (e.g. inviting others to join), or discouraging it in contexts of possible intergroup conflict (e.g. rapidly advertising alliance size in situations where alliance structure might otherwise be ambiguous). Moreover, individuals in groups might strategically (though often unconsciously) amplify aspects of colaughter to enhance the effect, and this could be implemented volitionally. Exaggerated emotional signals such as colaughter can potentially serve to provide unambiguous information about the constitution of a group and affect overhearers in adaptive ways for sender and receivers.

One limitation in the current study is the homogeneity in both our corpus of conversationalists from which the laughter was taken, as well as the experimental participants—both from WEIRD California university populations [72]. Consequently, even strangers in our conversations often have a fair amount of common ground, resulting in behaviours that signal friendly intentions. To a naive outsider, this could be difficult to discriminate from actual friends and thus result in high error rates and response biases like we see in the current study. Future research should examine speakers who are much more diverse on multiple demographic variables. The task of identifying friends and strangers would probably be much easier, and greater variability in the sample of conversationalists could afford a variety of predictions regarding how colaughter behaviour predicts compatibility. For example, perhaps newly acquainted strangers who immediately begin colaughing like familiar friends would cooperate more effectively and faster than those who do not engage in such a way. If so, a positive relationship between colaughing behaviour and actual cooperation between newly acquainted individuals could be an important reason why listeners are so attuned to the signal.

Empirical explorations of the communicative dynamics of colaughter in groups, the innumerable social functions it might serve, and its effects on subsequent group interactions, might reveal design features of a powerful species-specific signalling system that are only beginning to be understood.

Ethics. All procedures used for original data collection were approved by the UCLA Institutional Review Board (IRB#11-000928).

Data accessibility. All data and analysis scripts can be found at https://osf.io/7egry/.

Authors' contributions. G.A.B. and C.S.W conceived the study design, created the stimuli and collected the data; R.F. and G.A.B. conducted the data analysis; G.A.B., R.F. and C.S.W. wrote the manuscript.

Competing interests. The authors declare no competing interests.

Funding. The authors report no funding for this research.

Acknowledgements. We would like to thank Shaudee Salari and Amanda Schur for help running the experiments, and Andrey Anikin, Ray Gibbs, César Lima, Sam Mehr, Paul Smaldino, and an anonymous reviewer for comments on an earlier draft.

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
