## [Reviewer comments · Royal Society Open Science]

Review History

RSOS-201092.R0 (Original submission)

Review form: Reviewer 1 (Andrey Anikin)

Is the manuscript scientifically sound in its present form?

Yes

Are the interpretations and conclusions justified by the results?

Yes

Is the language acceptable?

Yes

Do you have any ethical concerns with this paper?

No

Have you any concerns about statistical analyses in this paper?

Yes

Recommendation?

Accept with minor revision (please list in comments)

Comments to the Author(s)

The subject of this study is the extent to which listeners can determine whether two interlocutors are friends or strangers. Following up on their earlier studies of this topic, the authors compared

the amount of information about affiliation available from laughter vs. speech in natural and sped-up versions. The main finding was that laughter provided more information about affiliation than did speech. The paper is clear and thoroughly researched; likewise, I find the experiment design solid and appropriate. Last but not least, the authors should be applauded for using cutting-edge statistical techniques and clearly putting a lot of effort into their data analysis. In sum, this is a highly competent piece of research with a clear message.

Accordingly, my comments are rather minor. The main theoretical issue that worries me is the lack of clarity about what psychological processes might be behind the differences in colughter between friends and strangers. The authors make a convincing case for potential evolutionary payoffs of advertising alliances, and the reasoning is very clear up to p. 8. Then, however, they point out that acoustically colughter between friends is indicative of high arousal, which they also link with spontaneous rather than volitional vocal production (p. 8, 1st par. - some more references could be helpful to back up this claim), and conclude that "colughter constitutes an intergroup signal of affiliation" (p. 20). So the implication appears to be that friends talking to each other are more genuinely amused (or whatever mental state is assumed to trigger laughter) than strangers or, in plain language, have more fun together? Or are they less inhibited / nervous (and if so, could their arousal levels be lower, not higher)? Does "friend-like" colughter tell observers that these two people like each other, feel relaxed in each other's company, are excited to be together, ...? Or are they exaggerating their amusement (which is not spontaneous) to advertise their coalition? For example, on p. 29 colughter is described as an "exaggerated emotional signal". It would be helpful if the authors spelled out their reasoning more clearly and clarified - or speculated - what proximal mechanisms might bridge evolutionary and cognitive levels of explanation.

Data availability: a link to OCF is provided, but the project is locked, so the only way to obtain the data is to contact the authors.

p. 4 "Here we define colughter as temporally coincident laughter" Perhaps a quick justification - why is temporally coincident laughter of particular theoretical interest?

Method, p. 9: "Listeners were asked to identify whether the pair of vocalizers were friends or strangers" When the task is to discriminate between two categories with a natural "high end" (friends = "more", strangers = "less"), the danger is that the same quantitative characteristics (high pitch, abrupt onset, etc.) may guide the responses, regardless of the precise nature of the task. For ex., a similar pattern is observed in other studies where participants rated authenticity, emotion intensity, valence, etc. Will the information about affiliation be salient to listeners in a real-life situation, when they are not explicitly asked to listen for it?

pp. 10-11 "Speakers were initially instructed to talk about bad roommate experiences... colughter segments were not different in duration ($M = 1.1$ s)" These appear to be rather short conversational chuckles (unfortunately, I haven't heard the audio). To what extent can we consider them spontaneous, compared to long and sometimes uncontrollable bouts of laughter when something truly amusing happens?

p. 7 "high energy voice-onset features (e.g., high pitch and loudness with abrupt onset) effective for grabbing attention" Reference?

p. 13 ""How much do you think these people liked each other?" Results for this second question are presented in the SM" The summary in the main text (p. 19) is useful but very short and easy to miss - maybe remind the readers what "Question 2" is?

Fig. 2 - why not show fitted values with CIs as well the the observed data? This might be more intuitive than Fig. 3, which is then not really needed.

p. 17 The model structure in Methods is clear, but here I don't know what is being compared. For ex., "a tendency to judge colaughter as between friends (mean difference 0.19, SD: 0.05..." Mean difference between what and what? And why suddenly "SD" instead of CI? Also: I take it that all differences in criterion and sensitivity are on the scale of the link function (probit)? How about reporting the differences on the scale of the outcome variable (proportions or %)? Are you comparing fitted values or regression coefficients (which shouldn't be meaningful in Model 4 with a triple interaction)? Finally, I'm not convinced that reporting results separately for criterion and sensitivity is the most intuitive approach in this case. Merely as a suggestion: you have 8 simple percentages in Fig. 2 - how about simply reporting differences between these fitted values (+double contrasts) as differences in % + 95% CI? You can still make all the same claims in the text, but much more transparently.

Section 3.1 seem like an overkill, and Fig. 5 is hard to read. Instead of comparing the variability of individual responses in different conditions (the table on p. 19), how about comparing the effect of conditions across participants (ie the variability in random slopes for each condition) to check whether there was more agreement among participants about the effects of speeding up vs. actual familiarity, etc? Just a thought.

p. 22 "overhearers can detect affiliation in overlapping talk – our results suggest that listeners can make this judgment better than chance" It would be useful to report the accuracy of judgments in Results.

Supplements

p. 1 "Data were modeled using a Gaussian link function with rating response (1 being "Not at all" and 7 being "very much") as a continuous function." Since you are using brms in the main text, it would be quite straightforward to do this analysis "properly" as well, using ordinal logistic regression instead of assuming that 1...7 ratings are continuous, equidistant, and normally distributed. See Liddell 2017 "Analyzing ordinal data with metric models" and Burkner 2019 "Ordinal regression models in psychology".

Tables S1 and S2 are not particularly helpful. How about just sticking with one model and reporting the relevant contrasts on the natural scale (1...7)?

Reviewed by Andrey Anikin

Review form: Reviewer 2

Is the manuscript scientifically sound in its present form?

Yes

Are the interpretations and conclusions justified by the results?

No

Is the language acceptable?

Yes

Do you have any ethical concerns with this paper?

No

Have you any concerns about statistical analyses in this paper?

No

Recommendation?

Major revision is needed (please make suggestions in comments)

Comments to the Author(s)

Bryant et al. present the results of a study on how well listeners are able to assess affiliation from colughter and cospeech. Listeners provided forced-choice judgments (friend vs stranger) and rated how much the two speakers liked each other. Despite the cospeech samples being over twice as long in duration as the colughter samples, listeners were less accurate when making judgements based on cospeech. Using sped-up versions of the stimuli did not change the overall accuracy but led listeners to more frequently judge pairs as friends. The authors thus conclude that we convey more information about affiliation through laughter than for other vocalisations, here cospeech. The study tackles an interesting question. It is written in a concise and informative way, the methods and procedure are described adequately and the statistics and interpretation of the results appear sound, although I am not an expert in the particular statistical analysis used. The literature review is thorough and results are well illustrated with informative figures.

I have reviewed this paper before at a different journal and most of my concerns have been addressed in a satisfactory manner. For example, in the previous review process, I was primarily concerned about using cospeech as an adequate baseline to tackle the research questions. I am copying the relevant paragraph from my previous review below:

“The selection criteria for cospeech seem to cover many different types of cospeech: back channelling behaviors, unsuccessful interruptions, successful turn taking, etc. The samples of cospeech provided by the authors in fact illustrate that there are different types of cospeech in their stimulus set. One example sounds like tightly timed turn taking, the other sounds like an unsuccessful interruption. This makes for a very messy category of speech samples with drastically different pragmatic functions and connotations: some instances of cospeech indicate successful sections of interactions (e.g. fast and precise turn taking) and others indicate potentially less successful sections of interactions (e.g. unsuccessful interruptions). It could be predicted that the type of cospeech will have a big effect on how well listeners can judge affiliation from the stimuli. Similarly, it is unclear whether the colughter stimuli are similarly heterogeneous or not.”

While I still think that the cospeech as a condition is very heterogeneous and there are questions marks over the adequacy of this condition as a baseline, the authors have revised their paper to more comprehensively describe the properties and nature of cospeech to address this concern satisfactorily.

Thus, only one concern about the interpretation of the findings remains. The authors claim in the abstract that “laughter can uniquely transmit rich information about social relationships to third party overhearers”. Similar statements can be found in other sections of the paper as well.

To my mind, this claim is too strong and not helpful. The current study only shows that affiliation can be more rapidly detected from instances of laughter compared to speech. Generalising the findings beyond these two vocalisations seems unwarranted as the current data cannot speak to whether colughter is indeed unique in signalling affiliations. I would encourage the authors to tone down these claims.

Decision letter (RSOS-201092.R0)

Dear Dr Bryant,

The editors assigned to your paper ("Recognizing affiliation in collaughter and cospeech") have now received comments from reviewers. We would like you to revise your paper in accordance with the referee and Associate Editor suggestions which can be found below (not including confidential reports to the Editor). Please note this decision does not guarantee eventual acceptance.

Please submit a copy of your revised paper before 12-Aug-2020. Please note that the revision deadline will expire at 00.00am on this date. If we do not hear from you within this time then it will be assumed that the paper has been withdrawn. In exceptional circumstances, extensions may be possible if agreed with the Editorial Office in advance. We do not allow multiple rounds of revision so we urge you to make every effort to fully address all of the comments at this stage. If deemed necessary by the Editors, your manuscript will be sent back to one or more of the original reviewers for assessment. If the original reviewers are not available, we may invite new reviewers.

- Data accessibility

If you wish to submit your supporting data or code to Dryad (<http://datadryad.org/>), or modify your current submission to dryad, please use the following link:
<http://datadryad.org/submit?journalID=RSOS&manu=RSOS-201092>

- Competing interests

- Authors' contributions

- Acknowledgements

- Funding statement

on behalf of Dr César Lima (Associate Editor) and Essi Viding (Subject Editor)
openscience@royalsociety.org

Comments to Author:

Reviewers' Comments to Author:

Reviewer: 1

Comments to the Author(s)

The subject of this study is the extent to which listeners can determine whether two interlocutors are friends or strangers. Following up on their earlier studies of this topic, the authors compared the amount of information about affiliation available from laughter vs. speech in natural and sped-up versions. The main finding was that laughter provided more information about affiliation than did speech. The paper is clear and thoroughly researched; likewise, I find the experiment design solid and appropriate. Last but not least, the authors should be applauded for using cutting-edge statistical techniques and clearly putting a lot of effort into their data analysis. In sum, this is a highly competent piece of research with a clear message.

Accordingly, my comments are rather minor. The main theoretical issue that worries me is the lack of clarity about what psychological processes might be behind the differences in colughter between friends and strangers. The authors make a convincing case for potential evolutionary payoffs of advertising alliances, and the reasoning is very clear up to p. 8. Then, however, they point out that acoustically colughter between friends is indicative of high arousal, which they also link with spontaneous rather than volitional vocal production (p. 8, 1st par. - some more references could be helpful to back up this claim), and conclude that “colughter constitutes an intergroup signal of affiliation” (p. 20). So the implication appears to be that friends talking to each other are more genuinely amused (or whatever mental state is assumed to trigger laughter) than strangers or, in plain language, have more fun together? Or are they less inhibited / nervous (and if so, could their arousal levels be lower, not higher)? Does “friend-like” colughter tell observers that these two people like each other, feel relaxed in each other’s company, are excited to be together, ...? Or are they exaggerating their amusement (which is not spontaneous) to advertise their coalition? For example, on p. 29 colughter is described as an “exaggerated emotional signal”. It would be helpful if the authors spelled out their reasoning more clearly and clarified – or speculated – what proximal mechanisms might bridge evolutionary and cognitive levels of explanation.

Data availability: a link to OCF is provided, but the project is locked, so the only way to obtain the data is to contact the authors.

p. 4 “Here we define colughter as temporally coincident laughter” Perhaps a quick justification – why is temporally coincident laughter of particular theoretical interest?

Method, p. 9: “Listeners were asked to identify whether the pair of vocalizers were friends or strangers” When the task is to discriminate between two categories with a natural “high end” (friends = “more”, strangers = “less”), the danger is that the same quantitative characteristics (high pitch, abrupt onset, etc.) may guide the responses, regardless of the precise nature of the task. For ex., a similar pattern is observed in other studies where participants rated authenticity, emotion intensity, valence, etc. Will the information about affiliation be salient to listeners in a real-life situation, when they are not explicitly asked to listen for it?

pp. 10-11 “Speakers were initially instructed to talk about bad roommate experiences... colughter segments were not different in duration ($M = 1.1$ s)” These appear to be rather short conversational chuckles (unfortunately, I haven’t heard the audio). To what extent can we consider them spontaneous, compared to long and sometimes uncontrollable bouts of laughter when something truly amusing happens?

p. 7 “high energy voice-onset features (e.g., high pitch and loudness with abrupt onset) effective for grabbing attention” Reference?

p. 13 ““How much do you think these people liked each other?” Results for this second question are presented in the SM” The summary in the main text (p. 19) is useful but very short and easy to miss – maybe remind the readers what “Question 2” is?

Fig. 2 – why not show fitted values with CIs as well the the observed data? This might be more intuitive than Fig. 3, which is then not really needed.

p. 17 The model structure in Methods is clear, but here I don’t know what is being compared. For ex., “a tendency to judge colughter as between friends (mean difference 0.19, SD: 0.05...” Mean difference between what and what? And why suddenly “SD” instead of CI? Also: I take it that all differences in criterion and sensitivity are on the scale of the link function (probit)? How about reporting the differences on the scale of the outcome variable (proportions or %)? Are you comparing fitted values or regression coefficients (which shouldn’t be meaningful in Model 4 with a triple interaction)? Finally, I’m not convinced that reporting results separately for criterion

and sensitivity is the most intuitive approach in this case. Merely as a suggestion: you have 8 simple percentages in Fig. 2 - how about simply reporting differences between these fitted values (+double contrasts) as differences in % + 95% CI? You can still make all the same claims in the text, but much more transparently.

Section 3.1 seem like an overkill, and Fig. 5 is hard to read. Instead of comparing the variability of individual responses in different conditions (the table on p. 19), how about comparing the effect of conditions across participants (ie the variability in random slopes for each condition) to check whether there was more agreement among participants about the effects of speeding up vs. actual familiarity, etc? Just a thought.

p. 22 “overhearers can detect affiliation in overlapping talk – our results suggest that listeners can make this judgment better than chance” It would be useful to report the accuracy of judgments in Results.

Supplements

p. 1 “Data were modeled using a Gaussian link function with rating response (1 being “Not at all” and 7 being “very much”) as a continuous function.” Since you are using brms in the main text, it would be quite straightforward to do this analysis “properly” as well, using ordinal logistic regression instead of assuming that 1...7 ratings are continuous, equidistant, and normally distributed. See Liddell 2017 “Analyzing ordinal data with metric models” and Burkner 2019 “Ordinal regression models in psychology”.

Tables S1 and S2 are not particularly helpful. How about just sticking with one model and reporting the relevant contrasts on the natural scale (1...7)?

Reviewed by Andrey Anikin

Reviewer: 2

Comments to the Author(s)

Bryant et al. present the results of a study on how well listeners are able to assess affiliation from colughter and cospeech. Listeners provided forced-choice judgments (friend vs stranger) and rated how much the two speakers liked each other. Despite the cospeech samples being over twice as long in duration as the colughter samples, listeners were less accurate when making judgements based on cospeech. Using sped-up versions of the stimuli did not change the overall accuracy but led listeners to more frequently judge pairs as friends. The authors thus conclude that we convey more information about affiliation through laughter than for other vocalisations, here cospeech. The study tackles an interesting question. It is written in a concise and informative way, the methods and procedure are described adequately and the statistics and interpretation of the results appear sound, although I am not an expert in the particular statistical analysis used. The literature review is thorough and results are well illustrated with informative figures.

I have reviewed this paper before at a different journal and most of my concerns have been addressed in a satisfactory manner. For example, in the previous review process, I was primarily concerned about using cospeech as an adequate baseline to tackle the research questions. I am copying the relevant paragraph from my previous review below:

“The selection criteria for cospeech seem to cover many different types of cospeech: back channelling behaviors, unsuccessful interruptions, successful turn taking, etc. The samples of cospeech provided by the authors in fact illustrate that there are different types of cospeech in their stimulus set. One example sounds like tightly timed turn taking, the other sounds like an unsuccessful interruption. This makes for a very messy category of speech samples with drastically different pragmatic functions and connotations: some instances of cospeech indicate successful sections of interactions (e.g. fast and precise turn taking) and others indicate

potentially less successful sections of interactions (e.g. unsuccessful interruptions). It could be predicted that the type of cospeech will have a big effect on how well listeners can judge affiliation from the stimuli. Similarly, it is unclear whether the colughter stimuli are similarly heterogeneous or not.”

While I still think that the cospeech as a condition is very heterogeneous and there are questions marks over the adequacy of this condition as a baseline, the authors have revised their paper to more comprehensively describe the properties and nature of cospeech to address this concern satisfactorily.

Thus, only one concern about the interpretation of the findings remains. The authors claim in the abstract that “laughter can uniquely transmit rich information about social relationships to third party overhearers”. Similar statements can be found in other sections of the paper as well.

To my mind, this claim is too strong and not helpful. The current study only shows that affiliation can be more rapidly detected from instances of laughter compared to speech. Generalising the findings beyond these two vocalisations seems unwarranted as the current data cannot speak to whether colughter is indeed unique in signalling affiliations. I would encourage the authors to tone down these claims.

Author's Response to Decision Letter for (RSOS-201092.R0)

See Appendix A.

RSOS-201092.R1 (Revision)

Review form: Reviewer 1 (Andrey Anikin)

Is the manuscript scientifically sound in its present form?

Yes

Are the interpretations and conclusions justified by the results?

Yes

Is the language acceptable?

Yes

Do you have any ethical concerns with this paper?

No

Have you any concerns about statistical analyses in this paper?

No

Recommendation?

Accept as is

Comments to the Author(s)

The authors have thoroughly revised the manuscript and addressed all my concerns. In particular, the Results section is much more accessible, and the supplements and R scripts on OSF

are extremely well-documented and super helpful. I have no reservations in recommending this manuscript for publication. Merely as a personal note (not a request for further revisions), you could also work with `brms::fitted()` instead of `posterior_samples` and thus obtain all your contrasts on the natural scale instead of probit, obviating the need to report both % and separate contrasts on the probit scale (eg in Table 1 - so it would simply be something like +13%, 95% CI [8%, 20%]).

Review form: Reviewer 2

Is the manuscript scientifically sound in its present form?

Yes

Are the interpretations and conclusions justified by the results?

Yes

Is the language acceptable?

Yes

Do you have any ethical concerns with this paper?

Yes

Have you any concerns about statistical analyses in this paper?

No

Recommendation?

Accept as is

Comments to the Author(s)

Thank you for making the changes in the revision. I can now accept this paper for publication. Congratulations!

Decision letter (RSOS-201092.R1)

Dear Dr Bryant,

It is a pleasure to accept your manuscript entitled "Recognizing affiliation in collaughter and cospeech" in its current form for publication in Royal Society Open Science. The comments of the reviewer(s) who reviewed your manuscript are included at the foot of this letter.

on behalf of Dr César Lima (Associate Editor) and Essi Viding (Subject Editor)
openscience@royalsociety.org

Reviewer comments to Author:
Reviewer: 2

Comments to the Author(s)
Thank you for making the changes in the revision. I can now accept this paper for publication. Congratulations!

Reviewer: 1

Comments to the Author(s)
The authors have thoroughly revised the manuscript and addressed all my concerns. In particular, the Results section is much more accessible, and the supplements and R scripts on OSF are extremely well-documented and super helpful. I have no reservations in recommending this manuscript for publication. Merely as a personal note (not a request for further revisions), you could also work with `brms::fitted()` instead of `posterior_samples` and thus obtain all your contrasts on the natural scale instead of probit, obviating the need to report both % and separate contrasts on the probit scale (eg in Table 1 - so it would simply be something like +13%, 95% CI [8%, 20%]).

Appendix A

UNIVERSITY OF CALIFORNIA, LOS ANGELES

UCLA

BERKELEY • DAVIS • IRVINE • LOS ANGELES • MERCED • RIVERSIDE • SAN DIEGO • SAN FRANCISCO

SANTA BARBARA • SANTA CRUZ

DEPARTMENT OF COMMUNICATION
2225 ROLFE HALL
BOX 951538
LOS ANGELES, CA 90095-1538

PHONE: (310) 825-3303
FAX: (310) 206-2371

August 13, 2020

Dear Drs. Lima and Essing,

Thank you very much for the opportunity to resubmit our manuscript “Detecting affiliation in colughter and cospeech” to *Royal Society Open Science*. We found the reviews very useful, and we have now addressed the comments from both reviewers. We believe the manuscript is notably improved. Below you will find our responses along with corresponding text we added or notes regarding where to find relevant changes in the revised manuscript.

We look forward to hearing back from you.

Best,

Greg Bryant
Christine Wang
Riccardo Fusaroli

Reviewers' Comments to Author:

Reviewer: 1

Comments to the Author(s)

The subject of this study is the extent to which listeners can determine whether two interlocutors are friends or strangers. Following up on their earlier studies of this topic, the authors compared the amount of information about affiliation available from laughter vs. speech in natural and sped-up versions. The main finding was that laughter provided more information about affiliation than did speech. The paper is clear and thoroughly researched; likewise, I find the experiment design solid and appropriate. Last but not least, the authors should be applauded for using cutting-edge statistical techniques and clearly putting a lot of effort into their data analysis. In sum, this is a highly competent piece of research with a clear message.

Thank you – this comment is very much appreciated.

Accordingly, my comments are rather minor. The main theoretical issue that worries me is the lack of clarity about what psychological processes might be behind the differences in colughter between friends and strangers. The authors make a convincing case for potential evolutionary payoffs of advertising alliances, and the reasoning is very clear up to p. 8. Then, however, they point out that acoustically colughter between friends is indicative of high arousal, which they also link with spontaneous rather than volitional vocal production (p. 8, 1st par. - some more references could be helpful to back up this claim),

and conclude that “colaughter constitutes an intergroup signal of affiliation” (p. 20). So the implication appears to be that friends talking to each other are more genuinely amused (or whatever mental state is assumed to trigger laughter) than strangers or, in plain language, have more fun together? Or are they less inhibited / nervous (and if so, could their arousal levels be lower, not higher)? Does “friend-like” colaughter tell observers that these two people like each other, feel relaxed in each other’s company, are excited to be together, ...? Or are they exaggerating their amusement (which is not spontaneous) to advertise their coalition? For example, on p. 24 colaughter is described as an “exaggerated emotional signal.” It would be helpful if the authors spelled out their reasoning more clearly and clarified – or speculated – what proximal mechanisms might bridge evolutionary and cognitive levels of explanation.

Thank you for pointing out the need for clarification on these points.

First off, we provided a couple of references regarding arousal and spontaneous vocal (and laugh) production, including one to the reviewer’s (and editor’s) work that should have been there in the first place (p. 8).

Regarding the main point, these particular stimuli were acoustically analyzed in our prior work (Bryant et al., 2016), and the implications of that analysis were that friends’ colaughter had relatively more features associated with speaker arousal and listeners used these features in making their judgments. We are confident in this part of the story.

But the reviewer raises an interesting issue concerning possible other scenarios, including reduction in inhibition and nervousness that could have other detectable acoustic effects. Ultimately, we suspect that many factors are likely at play in different contexts, and it results in familiar speakers being more amused, comfortable, and willing to signal with aroused enthusiasm, which of course is detectable through colaughter dynamics. We have added text to address this point (p. 8).

Other types of spontaneous emotional vocalizations differ from their volitional counterparts on similar acoustic features tied to speaker arousal, such as raised and more variable pitch, and lower harmonicity (Anikin & Lima, 2016). Sensitivity to indicators of speaker arousal could help overhearers track emotional engagement between interactants. This provides the basis for possible subsequent positive selection on groups of senders to amplify the signal, resulting in a ritualization process (Tinbergen, 1952). Proximately, this can manifest itself in familiar group members as heightened experiences of mirth and joy that motivates further colaughter. Moreover, there could be additional elements of group interactions that contribute to people’s affective experiences and resulting colaughter features. For example, reduced behavioral inhibition and increased overall comfort between familiar speakers could afford colaughing episodes that are not necessarily reflecting greater arousal, but more honest sounding vocal emotions and relaxed engagement. While recent studies strongly suggest arousal-linked qualities are playing a role in people’s judgments of familiarity, affiliation is potentially revealed in a variety of ways through interactive dynamics.

Finally, the issue about volitional exaggeration is also a great point, and could certainly be playing a role. We now elaborate on this (p. 24).

Moreover, individuals in groups might strategically (though unconsciously) amplify aspects of colaughter to enhance the effect, and this could be implemented volitionally.

We hope these additions adequately clarify our reasoning.

Data availability: a link to OCF is provided, but the project is locked, so the only way to obtain the data is to contact the authors.

Our apologies. This issue has been resolved (project is public) and the new zipped folder contains the updated R scripts.

p. 4 “Here we define colaughter as temporally coincident laughter” Perhaps a quick justification – why is temporally coincident laughter of particular theoretical interest?

We have added two sentences to better segue into the section that details the theoretical issues. Too much here will be redundant we feel (p. 4).

When people laugh in groups they often do so together, but limited work has explored how people laughing together affects perceivers or what information it might contain.

The focus of the current study is the perception of colaughter.

Method, p. 9: “Listeners were asked to identify whether the pair of vocalizers were friends or strangers” When the task is to discriminate between two categories with a natural “high end” (friends = “more”, strangers = “less”), the danger is that the same quantitative characteristics (high pitch, abrupt onset, etc.) may guide the responses, regardless of the precise nature of the task. For ex., a similar pattern is observed in other studies where participants rated authenticity, emotion intensity, valence, etc. Will the information about affiliation be salient to listeners in a real-life situation, when they are not explicitly asked to listen for it?

Important point that is a problem faced by most perception researchers who use rating scales or other language-based DVs. The basic finding that people discriminate friends from strangers in these stimuli has been shown in 24 societies, including several small scale cultures with participants having virtually no experience with perception experiments. We think this constitutes reasonable evidence that something about the colaughter is informing perceivers about affiliation that is well beyond more simple associations. It is also not clear to us that friends and strangers would necessarily map onto “more” and “less” in many of these societies, but that is an empirical question. Moreover, we don’t find effects for mean pitch which you might expect if a simple mapping was occurring. One virtue of our method, unlike many studies of this kind, is that the task has an actual correct answer (pairs were genuinely friends or strangers). The problem you describe, we believe, is more of an issue when judgments are essentially opinions, such as affective judgments and most rating scales. In any case, the issue is always something to consider.

pp. 10-11 “Speakers were initially instructed to talk about bad roommate experiences... colaughter segments were not different in duration ($M = 1.1$ s)” These appear to be rather short conversational chuckles (unfortunately, I haven’t heard the audio). To what extent can we consider them spontaneous, compared to long and sometimes uncontrollable bouts of laughter when something truly amusing happens?

We have now included several audio examples of the stimuli for reference. The basic answer is that some of the laughs are likely to be truly spontaneous, others volitional, and the distribution of these types is most likely different across friends and strangers, thus affording some discrimination by naïve listeners. None of the laughs are from long uncontrollable bouts—they

are all conversational and all are fairly low in overall arousal making our finding even more provocative we feel.

p. 7 “high energy voice-onset features (e.g., high pitch and loudness with abrupt onset) effective for grabbing attention” Reference?

We have provided two relevant references. One describes similar features of infant-directed speech (Fernald, 1992) and the other more generally describes this component in a form-function overview (Owren & Rendall, 2001).

p. 13 ““How much do you think these people liked each other?” Results for this second question are presented in the SM” The summary in the main text (p. 19) is useful but very short and easy to miss – maybe remind the readers what “Question 2” is?

Good suggestion. Done (p. 21).

Fig. 2 – why not show fitted values with CIs as well the observed data? This might be more intuitive than Fig. 3, which is then not really needed.

See below.

p. 17 The model structure in Methods is clear, but here I don’t know what is being compared. For ex., “a tendency to judge collaughter as between friends (mean difference 0.19, SD: 0.05...” Mean difference between what and what? And why suddenly “SD” instead of CI? Also: I take it that all differences in criterion and sensitivity are on the scale of the link function (probit)? How about reporting the differences on the scale of the outcome variable (proportions or %)? Are you comparing fitted values or regression coefficients (which shouldn’t be meaningful in Model 4 with a triple interaction)? Finally, I’m not convinced that reporting results separately for criterion and sensitivity is the most intuitive approach in this case. Merely as a suggestion: you have 8 simple percentages in Fig. 2 - how about simply reporting differences between these fitted values (+double contrasts) as differences in % + 95% CI? You can still make all the same claims in the text, but much more transparently.

We agree with the reviewer’s suggestions to improve the clarity of the results by following more directly on our hypotheses instead of on the estimates of the model.

We are now more explicitly stating in both methods and results that criterion and sensitivity per each condition (Table 1 and Figure 3) and their differences (text in the results section) were calculated ad hoc from the model posterior estimates (“fitted values”). We also explicitly state that the values are reported on a probit scale (z-scores).

We have corrected the 2 reported SDs to CIs (an oversight for which we apologize).

We understand the reviewer’s concerns with the difficulty of interpreting z-scores compared to probabilities (which motivated our Figure 2), and with the distinction between criterion and sensitivity. We have therefore thoroughly rewritten our results section. We keep Table 1 with the criterion, sensitivity and AUC per each condition, as these were the parameters explicitly estimated by the SDT model, and facilitate both error assessment (i.e. false positives and

negatives) and comparison with similar SDT studies. Accuracy estimates are thus covered by AUC, but we have also added estimates of % of “friends” response to the Table to facilitate interpretation.

Further, we have thoroughly revised the text in the results section to both follow more explicitly from our hypotheses and include percentage estimates:

As hypothesized, participants were able to judge colaughter and cospeech above chance ($AUC > .5$, see Table 1), but were credibly better at judging colaughter: the difference in sensitivity between conditions was 0.37 (on a z-score scale), 95% CIs: 0.24 0.51, 100% credibility, that is, 100% of the estimated parameter values indicate a higher sensitivity for colaughter. In particular, colaughter was judged accurately about 9% more often than cospeech. More exploratorily we can observe that participants were more likely to judge as produced by friends colaughter (62% of responses) than cospeech (47.5% of responses), a pattern that holds also for stimuli produced by strangers only, with a false positive rate of 41% for cospeech and 49% for colaughter (mean difference in criterion between colaughter and cospeech: -0.19, 95% CIs: 0.07 0.32, 99.97% credibility). The overall pattern indicates that cospeech tends to generate more false negatives (cospeech produced by friends judged as produced by strangers) and colaughter more false positives (colaughter produced by strangers judged as produced by friends), but the total error rate is higher in cospeech.

As hypothesized, speeding up colaughter increases attribution of the stimuli to friends: 68% of “friends” responses, against 62% in the original non-spaced-up colaughter. The effect is due to participants developing a positive bias for colaughter produced by strangers: the “friends” responses for the latter went from 49% to 56%, increase in criterion of 0.19, 95% CIs: 0.09 0.29, 100% credibility. On the contrary, their ability to judge colaughter produced by friends did not credibly change (going from 77% of “friends” responses to 80%, mean difference in sensitivity between original and spaced-up colaughter: -0.01, 95% CIs: -0.14 0.16, 33% credibility). However, against our predictions, the effects of speeding up were not credibly different between colaughter and cospeech (mean difference in criterion: -0.02, 95% CIs: -0.16 0.11, 63.6% credibility; mean difference in sensitivity: 0.1, 95% CIs: -0.12 0.31, 84.6% credibility). Speeding up cospeech did affect participants’ judgments by increasing their false positives rate from 41% to 50% (increase in criterion for spaced up compared to original cospeech: 0.22, 95% CI 0.12 0.31) and decreasing their sensitivity (-0.11, 95% CIs: -0.24 0.03, 94.73% credibility).

Section 3.1 seem like an overkill, and Fig. 5 is hard to read. Instead of comparing the variability of individual responses in different conditions (the table on p. 19), how about comparing the effect of conditions across participants (ie the variability in random slopes for each condition) to check whether there was more agreement among participants about the effects of speeding up vs. actual familiarity, etc? Just a thought.

We have now thoroughly revised section 3.1, analogously to the previous results section and moved it to the supplementary materials to accommodate the reviewers’ concerns. We now directly visualize effects of colaughter vs. cospeech and of the speed manipulation within single participants, quantifying how many participants display the population level effect and their individual variability. Given the extensive nature of the revisions, instead of pasting here the full section we refer to the manuscript.

p. 22 “overhearers can detect affiliation in overlapping talk—our results suggest that listeners can make this judgment better than chance” It would be useful to report the accuracy of judgments in Results.

We agree with the reviewer, and we have now more explicitly reported accuracies in the text, as well as more explicitly pointed to AUC in Table 4 and Figure 4 as a measure of accuracy. See revised Results text above.

Supplements

p. 1 “Data were modeled using a Gaussian link function with rating response (1 being “Not at all” and 7 being “very much”) as a continuous function.” Since you are using brms in the main text, it would be quite straightforward to do this analysis “properly” as well, using ordinal logistic regression instead of assuming that 1...7 ratings are continuous, equidistant, and normally distributed. See Liddell 2017 “Analyzing ordinal data with metric models” and Burkner 2019 “Ordinal regression models in psychology”.

The models in the supplementary materials (previously linear lme4 models) have now been replaced with ordinal Bayesian models and accordingly reported. The findings are the same as previously reported.

Tables S1 and S2 are not particularly helpful. How about just sticking with one model and reporting the relevant contrasts on the natural scale (1...7)?

We have now removed former Table S1 and provide a more detailed analysis of the findings. For what regards Table S2, we adopted a model comparison approach (together with regularizing priors) to avoid overfitting the data on our main results. We opted, for transparency, to keep Table S2 in the supplementary materials, as it doesn’t disrupt the flow of the manuscript and adds additional information for the interested reader.

Reviewed by Andrey Anikin

Thank you Andrey, your comments were very helpful!

Reviewer: 2

Comments to the Author(s)

Bryant et al. present the results of a study on how well listeners are able to assess affiliation from laughter and cospeech. Listeners provided forced-choice judgments (friend vs stranger) and rated how much the two speakers liked each other. Despite the cospeech samples being over twice as long in duration as the laughter samples, listeners were less accurate when making judgements based on cospeech. Using sped-up versions of the stimuli did not change the overall accuracy but led listeners to more frequently judge pairs as friends. The authors thus conclude that we convey more information about affiliation through laughter than for other vocalisations, here cospeech. The study tackles an interesting question. It is written in a concise and informative way, the methods and procedure are described adequately and the statistics and interpretation of the results appear sound, although I am not an expert in the particular statistical analysis used. The literature review is thorough and results are well illustrated with informative figures.

I have reviewed this paper before at a different journal and most of my concerns have been addressed in a satisfactory manner. For example, in the previous review process, I was primarily concerned about using cospeech as an adequate baseline to tackle the research questions. I am copying the relevant paragraph from my previous review below:

“The selection criteria for cospeech seem to cover many different types of cospeech: back channelling behaviors, unsuccessful interruptions, successful turn taking, etc. The samples of cospeech provided by the authors in fact illustrate that there are different types of cospeech in their stimulus set. One example sounds like tightly timed turn taking, the other sounds like an unsuccessful interruption. This makes for a very messy category of speech samples with drastically different pragmatic functions and connotations: some instances of cospeech indicate successful sections of interactions (e.g. fast and precise turn taking) and others indicate potentially less successful sections of interactions (e.g. unsuccessful interruptions). It could be predicted that the type of cospeech will have a big effect on how well listeners can judge affiliation from the stimuli. Similarly, it is unclear whether the colughter stimuli are similarly heterogeneous or not.”

While I still think that the cospeech as a condition is very heterogeneous and there are questions marks over the adequacy of this condition as a baseline, the authors have revised their paper to more comprehensively describe the properties and nature of cospeech to address this concern satisfactorily.

Thus, only one concern about the interpretation of the findings remains. The authors claim in the abstract that “laughter can uniquely transmit rich information about social relationships to third party overhearers”. Similar statements can be found in other sections of the paper as well.

To my mind, this claim is too strong and not helpful. The current study only shows that affiliation can be more rapidly detected from instances of laughter compared to speech. Generalising the findings beyond these two vocalisations seems unwarranted as the current data cannot speak to whether colughter is indeed unique in signalling affiliations. I would encourage the authors to tone down these claims.

Thank you very much for your helpful earlier review, and these comments. We agree that claims of uniqueness are not strong when colughter is only compared to one other vocal mode. We changed the wording in the abstract (removed “uniquely”) and altered the stated aims of the study by claiming we were investigating “uniqueness” and instead now say “relative efficacy” (p. 9). We found no other places where the uniqueness claim is made.